# Potential Molecular Mechanism of *Illicium simonsii* Maxim Petroleum Ether Fraction in the Treatment of Hepatocellular Carcinoma

**DOI:** 10.3390/ph17060806

**Published:** 2024-06-19

**Authors:** Sihua Zou, Yanchun Wu, Meiqi Wen, Jiao Liu, Minghui Chen, Jingquan Yuan, Bei Zhou

**Affiliations:** 1School of Pharmacy, Guangxi University of Chinese Medicine, Nanning 530200, China; positive1234567@163.com (S.Z.); jiaoentry@163.com (J.L.); 15677123776@163.com (M.C.); 2Guangxi Scientific Research Centre of Traditional Chinese Medicine, Guangxi University of Chinese Medicine, Nanning 530200, China; wuyanchun_3333@126.com (Y.W.); wmq0073@163.com (M.W.); 3Guangxi Key Laboratory of Efficacy Study on Chinese Materia Medica, Nanning 530200, China

**Keywords:** petroleum ether fraction of *Illicium simonsii* Maxim, hepatocellular carcinoma, mechanism

## Abstract

Traditional Chinese medicine (TCM) has been considered, for many years, an important source of medicine to treat different diseases. As a type of TCM, *Illicium simonsii* Maxim (ISM) is used as an anti-inflammatory, anti-bacterial, and anti-virus. Besides, ISM is also used in the treatment of cancer. In order to evaluate the anti-hepatocellular carcinoma (HCC) activity, petroleum ether extract was prepared from part of the fruit of ISM. First, the compounds of the petroleum ether fraction of *Illicium simonsii* Maxim (PEIM) were identified using LC-MS/MS analysis. Next, the cell viability and morphological changes were evaluated by MTT assay and Hoechst staining. In addition, the effect of PEIM on the levels of inflammatory factors (TNF-α, IL-1β, and IL-6) was determined using the ELISA kit. Furthermore, apoptosis was evaluated by flow cytometry, and gene expression and the regulation of signaling pathways were investigated, respectively, by real-time fluorescence quantitative PCR (RT-qPCR) and western blot. Results showed that a total of 64 compounds were identified in the PEIM. Additionally, the PEIM had anti-HCC activity against HepG2 cells, in which the half maximal inhibitory concentration (IC_50_) was 55.03 μg·mL^−1^. As well, the PEIM was able to modulate the expression of TNF-α, IL-1β, and IL-6, while we also found that it induced HepG2 cell apoptosis through the activation of P53 mRNA and caspase-3 mRNA. Finally, the PEIM possibly downregulated the expression of TLR4, MyD88, p-NF-κBp65, TNF-α, IL-1β, INOS, IL-6, JAK2, STAT3, CyclinD1, CDK4, MDM2, and Bcl-2, and upregulated the expression of P53, P21, Bax, Cytochrome-C, Caspase-9, and Caspase-3 in HepG2 cells. These findings may confirm that the PEIM has possible anti-HCC effects. However, additional studies are required to fully understand the mechanisms of action of the PEIM and the signaling pathways involved in its effects. Moreover, the anti-HCC activity of the PEIM should be studied in vivo, and signaling pathways involved in its effects should be explored to develop the anti-HCC drug.

## 1. Introduction

HCC, an aggressive malignant disease, is commonly known as the king of cancer and one of the most common malignant tumors in the clinic, and the 5-year survival rate is only 18% in the United States (USA), while it is only 12% in China [1,2,3]. HCC belongs to the category of liver accumulation and abdominal mass in traditional Chinese medicine science, that is, a syndrome of deficiency in origin and excess in superficiality [4]. Modern medicine believes that long-term heavy drinking, eating moldy food, genetic and environmental factors, and so on may be the causes of HCC. Seventy percent of HCC is already in the intermediate and advanced stages once discovered [5]. Interventional therapy, targeted therapy (sorafenib or lenvatinib, etc.) and immunotherapy (PD-1/PD-L1 antibodies, etc.), are often used for intermediate and advanced HCC [6]. However, both targeted therapy and immunotherapy have disadvantages. On one hand, targeted therapy is prone to drug resistance and has a high cost. On the other hand, immunotherapy has side effects such as edema, fever, and skin toxicity. In order to overcome or reduce their side effects, it is necessary to develop an alternative chemotherapeutic or complementary strategy to treat HCC. Hence, TCM and its bioactive compounds can be explored as safer alternatives in the form of combinational treatment strategies in addition to interventional therapy, targeted therapy, and immunotherapy. It is worth mentioning that toxic TCM has been proven to be able to treat HCC, which may be related to the toxicity. However, studies have shown that toxic TCM may play a role in anti-HCC by regulating a variety of signaling pathways, such as NF-κB and STAT3 [7,8].

*Illicium* L. has 40 species and a long history of medicinal use in the world. It was recorded in the Compendium of Materia Medica more than 400 years ago. Besides, *Illicium* L. is a source of sesquiterpenes, diterpenes and triterpenoids, flavonoids, phenylpropanoids, lignans, and volatile oils [9]. ISM, a member of *Illicium* L., is mainly indigenous in India, Myanmar, and the southwestern part of China [10]. In addition, the research on ISM mainly focuses on the separation of its chemical compounds and pharmacological activities, such as anti-inflammatory [11], antibacterial [12], and neuroprotective effects [13]. In addition, anisatin and (1S)-minwanenone separated from ISM can inhibit the growth of SMMC-7721 cells [14]. Our previous study showed that the DEIM (dichloromethane fraction of *Illicium simonsii* Maxim) and PEIM have an anti-HCC effect. In particular, the PEIM has the best anti-HCC effect on HepG2 cells (IC_50_ = 55.03 μg·mL^−1^) and Bel-7404 cells (IC_50_ = 59.67 μg·mL^−1^). The inhibitory effect of the PEIM on HepG2 cells was better than that on Bel-7404 cells, so we chose HepG2 cells for further study. Significant scientific evidence shows the importance of TCM in the development of new drugs to treat HCC, which is a major disease that threatens human life and health, so research into anti-HCC drugs can’t wait. The current study aims to assess the potential beneficial effect of the PEIM in HCC therapy. We hypothesized that the PEIM would potentiate the anticancer effect by inhibiting cell proliferation, promoting cell death, and modulating the signaling pathways associated with the development of HCC, and that it will contribute to improving the quality of life of HCC patients and unraveling a new treatment strategy that is more effective with fewer or no side effects.

## 2. Results

### 2.1. LC-MS/MS Analyzed the Chemical Substances of PEIM

The chemical substances in the PEIM were analyzed by the LC-MS/MS analysis method. Compounds were identified by correlating the molecular ion peaks with MS fragmentation to that reported by previous researchers or online software programs, which are represented in Table 1, that displays the retention time (tR), *m*/*z* found in both modes (ESI^+^/ESI^−^), and a fragmentation pattern, along with the actual *m*/*z* of the identified compounds. A total of 64 compounds were tentatively identified in the PEIM by comparing the database with the literature. These compounds are listed in Table 1. Flavonoids, diterpenoids, triterpenoids, alkaloids, lignans, and sesquiterpenoids were the identified classes of constituents in the PEIM. Flavonoids were the major class of the identified active constituents. Compounds with anti-HCC effects include glabridin [15], baohuoside I [16], kaempferol [17], isorhamnetin [18], morin hydrate [19], kaempferitrin [20], genistein [21], pectolinarigenin [22], quercetin [23], luteolin [24], tangeretin [25], calycosin [26], and hyperin [27]. The total ion mass spectrometry is shown in Figure 1, and the structure of the most abundant flavonoid in the PEIM is shown in Figure 2.

### 2.2. PEIM Affected the Proliferation and Morphology in HepG2 Cells

The MTT method was used to analyze the effect of the PEIM on the proliferation of HepG2 cells after 48 h (Figure 3A). Hoechst33342 staining was used to observe the morphology of apoptosis after the HepG2 cells were treated with the PEIM. In this experiment, with the increase in drug concentration, the amount of cell debris increased, and the apoptotic cells showed a clear, bright blue, round or condensed, clumpy structure (Figure 3B).

### 2.3. PEIM Decreased the Content of TNF-α, IL-1β and IL-6 in HepG2 Cells

ELISA assay was used to analyze the effect of the PEIM on the content of TNF-α, IL-1β, and IL-6 in the cell supernatant. After treatment with different concentrations of the PEIM, the contents of TNF-α (Figure 4A), IL-1β (Figure 4B) and IL-6 (Figure 4C) in the supernatant of HepG2 cells decreased gradually compared to the control group (*p* < 0.05 or *p* < 0.01).

### 2.4. PEIM Promoted Apoptosis in HepG2 Cells

The apoptotic effect of the PEIM was dose-dependent. Whether the PEIM inhibited HepG2 cells through apoptosis was further determined by flow cytometry. As shown in Figure 4B and Figure 5A, the number of late apoptotic and total apoptotic HepG cells increased gradually after treatment with the PEIM (*p* < 0.05).

### 2.5. PEIM Increased the Expression of P53 mRNA and Caspase-3 mRNA in HepG2 Cells

RT-qPCR was used to analyze the effect of the PEIM on the expression of P53 mRNA and caspase-3 mRNA from a genetic perspective. The expression of P53 mRNA (Figure 6A) and caspase-3 mRNA (Figure 6B) in HepG2 cells was gradually increased, and showed a discernible dose-dependent pattern. Notably, the expression of p53 mRNA and caspase-3 mRNA showed a significant difference (*p* < 0.05) compared to the control group after treatment with different concentrations of 50 μg·mL^−1^ PEIM. 

### 2.6. PEIM Affected the Expression of TLR4/MyD88/NF-κB, JAK2/STAT3, P53/P21/MDM2, and Mitochondrial Apoptosis Pathway-Related Proteins in HepG2 Cells

In order to further study the mechanism of the PEIM against HCC, Western blot assay was used to detect the expression of TLR4/MyD88/NF-κB, JAK2/STAT3, P53/P21/MDM2, and mitochondrial apoptosis pathway-related proteins. First, the results showed that the PEIM could down-regulate the expression of TLR4, MyD88, p-NF-κB p65, TNF-α, IL-1β, and INOS (Figure 7A). Second, the expression of IL-6, JAK2, and STAT3 were also down-regulated (Figure 7B). Third, we found that the expression of P53 and P21 were up-regulated, and the expression of CDK4, CyclinD1, and MDM2 were down-regulated (Figure 7C). Finally, the expression of Bax, cytochrome-C, caspase-9, and caspase-3 were up-regulated, and the expression of Bcl-2 protein was down-regulated (Figure 7D). 

## 3. Discussion

This study analyzed the chemical constituents of the PEIM by LC-MS/MS and investigated the effect of the PEIM against HCC, showing that a total of 64 compounds were identified, including flavonoids, diterpenoids, triterpenoids, alkaloids, lignans, and sesquiterpenoids, which are believed to play a crucial role in treating HCC. 

Based on the results of the MTT assay study, the PEIM showed cytotoxicity against HepG2 cells, and the IC_50_ value was 55.03 μg·mL^−1^. Moreover, the results also showed, by Hoechst33342 staining assay, that the PEIM could cause HepG2 cell lysis and death, which was used to verify whether the PEIM could affect the morphology of HepG2 cells.

It is well known that promoting tumor inflammation is one of the most important characteristics of malignant tumors and plays an important role in the development, invasion, and metastasis of tumors [28]. In addition, it is generally believed that TLR4/NF-κB/MyD88 and JAK2/STAT3 are the important signaling pathways related to inflammation [29,30]. For example, it has been reported that the TLR4/NF-κB/MyD88 signaling pathway can induce the transcription and expression of a variety of pro-inflammatory chemokines, such as IFN-γ and TNF-α, related to liver inflammation [31]. Additionally, down-regulation of IL-6, JAK2, and STAT3 could inhibit cell proliferation in HepG2 cells [32,33]. Therefore, the expression of proteins related to the TLR4/NF-κB/MyD88 and JAK2/STAT3 signaling pathways was also assessed by Western blot, and the content of inflammatory factors (TNF-α, IL-1β, and IL-6) was assessed using an ELISA kit. The results of this study demonstrate that the PEIM potentiates the anti-HCC effect by reducing inflammation due to down-regulating the expression of TLR4, MyD88, p-NF-κBp65, TNF-α, IL-1β, INOS, IL-6, JAK2, and STAT3, and reducing the content of TNF-α, IL-1β, and IL-6. It was noteworthy that the above-mentioned proteins showed significant differences when the concentration was at 50 μg·mL^−1^. Based on the above results, we believe that the PEIM may play an anti-HCC role from the perspective of increasing immunity.

Apoptosis inhibition is known to be one of the means by which cancer cells assure proliferation and survival. Furthermore, it is well known that P53/P21/MDM2 and the mitochondrial signaling pathway play a significant role in apoptosis. On one hand, p53 is a tumor suppressor gene and the upstream factor of P21, which can induce apoptosis via the inhibition of cyclinD1 and CDK4 of HCC cells [34,35]. On the other hand, Bcl-2 plays a tumor suppressor role in the mitochondrial apoptosis pathway by blocking the pro-apoptotic effect of Bax. Additionally, it will induce the release of cytochrome-C when the integrity of the mitochondria is destroyed, and then the expression of caspase proteins will be activated [36]. To further understand the mechanism by which the PEIM exerts its effects, apoptosis was detected by flow cytometry, showing that the PEIM induced apoptosis of HepG2 cells, and the total apoptosis rate was 75.5% at a concentration of 50 μg·mL^−1^. RT-qPCR results showed that the PEIM could promote the relative expression of P53 mRNA and caspase-3 mRNA, which was a noteworthy finding from our study. Moreover, Western blot results showed that the PEIM could induce apoptosis of HepG2 cells due to upregulation of P53, P21, cytochrome-C, caspase-9, and caspase-3, as well as downregulation of cyclinD1, CDK, MDM2, and Bcl-2 expression. It is noteworthy that caspase-9 and caspase-3 showed extremely significant differences when the concentration was at 50 μg·mL^−1^. We strongly believe that the development of strategies aimed at enhancing cell death will open the prospect of improving the success of cancer treatment by combining these natural TCM therapies with conventional chemotherapies. The summary figure for each pathway investigated is shown in Figure 8.

## 4. Materials and Methods

### 4.1. Drug Preparation 

The ISM in this study was collected from Tongren, Guizhou. It was identified by Professor Ma Wenfang of Guangxi University of Traditional Chinese Medicine. The HepG2 cell directory number was TCHu 72, and they were purchased from the cell bank of the Chinese Academy of Sciences. After crushing, the medicinal materials were extracted three times with 95% ethanol by micro-boiling reflux for 2 h each time. The obtained liquid was concentrated by a rotary evaporator. After the extract was obtained, water and petroleum ether were added for extraction. Finally, the PEIM was obtained after concentration.

### 4.2. Liquid Chromatography-Mass Spectroscopy Study

A UPLC-Q Exactive quadrupole-electrostatic field orbitrap high-resolution mass spectrometer (Thermo Fisher Scientific, Waltham, MA, USA) equipped with a HESI source was used to detect the chemical components of the PEIM. A volume of 3 μL of the sample was injected, and mass spectra were investigated in the range of 50–1000 Da by applying negative as well as positive ionization modes, where the spray voltage was 3.5 kV (+) and 3.2 kV (−), sheath gas volume flow was 30 μL·min^−1^, ion transport tube temperature was 320 °C, auxiliary gas flow rate was 10 μL·min^−1^, and auxiliary gas temperature was 300 °C. To conduct the HPLC analysis, a Waters Alliance 2695 HPLC Pump (Waters, Milford, MA, USA) was employed along with a Thermo Gold C18 column, of which the specification was 2.1 mm × 100 mm, 1.9 μm (Thermo Fisher Scientific, Waltham, MA, USA). The mobile phase comprised (A) 0.1% formic acid (95.0) (Fisher, Hampton, NH, USA) and (B) acetonitrile (5.0) (Fisher, Hampton, NH, USA). The gradient condition was: 0–20 min, gradient from 5% of B; 20–23 min, isocratic conditions at 95% of B; 23–24 min, gradient from 95% of B; 24–27 min, isocratic conditions at 5% of B. Flow rate: 0.4 mL·min^−1^. The identification of components was accomplished.

### 4.3. Cell Culture and Treatment

HepG2 cells were purchased from the Cell Bank of Chinese Academy of Sciences, and grown in the medium containing 10% DMEM (Gibco/Thermo Fisher Scientific, Waltham, MA, USA) and 1% FBS (VivaCell, Shanghai, China) at 37 °C in a cell incubator (Sanyo, Osaka, Japan). We observed cell morphology with an inverted microscope (Olympus Corporation, Tokyo, Japan) or fluorescence microscope (Carl Zeiss, Oberkochen, Germany).

### 4.4. Cell Viability Assay

An MTT assay was used to test the effect of the PEIM on the proliferation of HepG2 cells. MTT was diluted with phosphate-buffered saline (PBS) to a 5 mg·mL^−1^ solution. HepG2 cells were seeded in 96-well plates at a density of 1 × 10^4^ cells/well, and then the plates were placed in a cell incubator and incubated at 37 °C with 5% CO_2_ for 24 h. PEIM was dissolved in dimethyl sulfoxide (DMSO), and different concentrations were set. In addition, the corresponding control group containing DMSO was set. Five duplicate wells were set for each concentration. After incubation for 24 h, 48 h, and 72 h, the supernatant was discarded, and 10 μL of 5 mg·mL^−1^ MTT solution was added to each well. After incubation for 4 h in the incubator, the supernatant was discarded, and 150 μL of dimethyl sulfoxide was added. The blue-purple formazan crystal was dissolved in a shaker for 10 min. Finally, the absorbance at 490 nm was detected by a microplate reader (Bio-Rad, Hercules, CA, USA).

### 4.5. Enzyme-Linked Immunosorbent Assay 

The cell supernatant was collected and centrifuged at 2–8 °C, 1000× *g* for 20 min to remove impurities and cell debris. The levels of TNF-α, IL-1β, and IL-6 in the cell supernatant were detected by an ELISA kit (Elabscience, Wuhan, China).

### 4.6. Hoechst33342 Staining

HepG2 cells were treated with different concentrations of the PEIM for 24 h, 200 μL of 10 Hoechst33342 staining (Solarbio, Beijing, China) was added at a concentration of 10 μg·mL^−1^ and incubated for 10 min. The cell morphology was observed under a fluorescence microscope (Zeiss, Oberkochen, Germany) and photographed.

### 4.7. FACS Analysis 

The HepG2 cells in the logarithmic growth phase were washed with PBS and digested with a 0.25% trypsin-EDTA digestive solution. The cells were collected by centrifugation, resuspended in PBS, and inoculated in 6-well plates at 1 × 10^6^/well in a cell incubator for 24 h. The old medium was discarded, and different concentrations (50 μg·mL^−1^, 25 μg·mL^−1^, and 12.5 μg·mL^−1^) of drugs were added for intervention. After 48 h, the old medium was discarded and the cells were collected. After being washed twice with PBS, 500 μL 1×Binding Buffer was added to re-suspend the cells, and 5 μL Annexin V-FITC staining solution and 10 μL propidium iodide (PI) staining solution were added at room temperature for 5 min, detected by flow cytometry (Becton Dickinson, Franklin Lakes, NJ, USA).

### 4.8. RT-qPCR Assay 

Trizol lysis buffer (TaKaRa, Kyoto, Japan) was used to lyse the cells. Carbon tetrachloride was added after standing at room temperature for 5 min. After shaking, the cells were allowed to stand at room temperature for 3 min, and then centrifuged at 4 °C and 12,000 rpm for 15 min. The upper water phase was transferred to the EP tube, and isopropanol with the same volume as water was added. After mixing, the mixture was allowed to stand at room temperature for 10 min, and then centrifuged at 4 °C at 12,000 rpm for 10 min. The supernatant was discarded, and 500 μL of 75% ethanol was added and centrifuged at 4 °C at 10,000 rpm for 5 min. The supernatant was discarded, and nuclease-free water was added to obtain the total RNA that was reverse transcribed into DNA using a reverse transcription kit (Thermo Fisher Scientific, Waltham, MA, USA) using a gradient PCR instrument (BIO-RAD, Hercules, CA, USA). Finally, a Roche 96 PCR instrument (Roche, Basel, Switzerland) was used for detection. The primers of RT-qPCR are shown in Table 2. 

### 4.9. Western Blot Analysis 

RIPA lysis buffer and a protease inhibitor were used to lyse the cells to extract the protein, and the protein concentration was determined. The extracted protein was separated by SDS-PAGE gel electrophoresis and transferred to the PVDF membrane. After blocking with a rapid blocking solution at room temperature for 10 min, the membranes were incubated with the primary antibodies at 4 °C overnight. After washing, the membranes were further incubated with secondary antibodies at room temperature for 1 h. Finally, the protein bands were visualized using an Omni-ECLTM ultrasensitive chemiluminescence detection kit (Epizyme, Shanghai, China) by an imaging system (Bio-Rad, Hercules, CA, USA). Primary antibodies for TLR4, MyD88, p-NF-κBp65, IL-1β, TNF-α, INOS, P53, MDM2, cyclinD1, CDK4, Bcl-2, Bax, caspase-9, and cytochrome-C were purchased from Proteintech (Wuhan, China). IL-6 was purchased from Beyotime (Shanghai, China). JAK2 and STAT3 were purchased from HUABIO (Hangzhou, China). P21 and caspase-3 were purchased from CST (Danvers, MA, USA).

### 4.10. Statistical Analysis

The results were analyzed by SPSS25.0 software and are shown as mean ± standard deviation. SPSS25.0 was used for one-way analysis of variance; Prism 8.0.2 and ImageJ 1.8.0 were used to analyze the statistical differences and draw. A *p* value of less than 0.05 was considered statistically significant.

## 5. Conclusions

In conclusion, the results of this study suggested that the PEIM showed a prominent anti-HCC activity and increased cell apoptosis in vitro. Such observations might be due to the presence of biologically functional compounds in the PEIM. Additionally, we demonstrated that the mechanism behind the effects of the PEIM is by acting on the TLR4/MyD88/NF-κB, IL-6/JAK2/STAT3, P53/P21/MDM2, and mitochondrial apoptosis pathways. These data suggested that PEIM could be a powerful new drug against HCC by modulating proliferation, cell death, and the above-mentioned signaling pathways. However, further studies are needed to better understand the molecular mechanisms and further investigate the signaling pathways involved in different HCC cells. 

## Figures and Tables

**Figure 1 pharmaceuticals-17-00806-f001:**
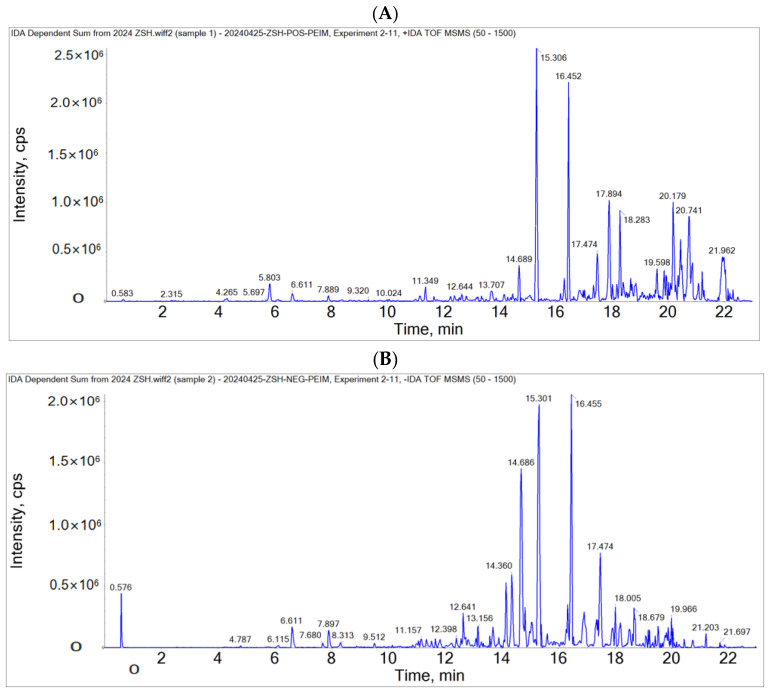
Total ion mass spectrometry of PEIM. (**A**) Total positive ion diagram. (**B**) Total negative ion diagram.

**Figure 2 pharmaceuticals-17-00806-f002:**
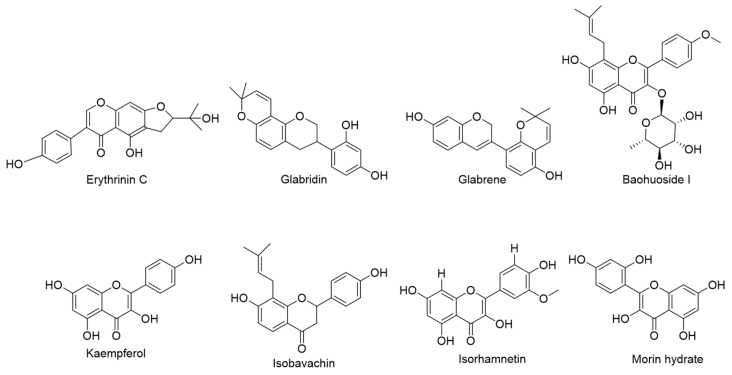
The structures of the most abundant flavonoids in the PEIM.

**Figure 3 pharmaceuticals-17-00806-f003:**
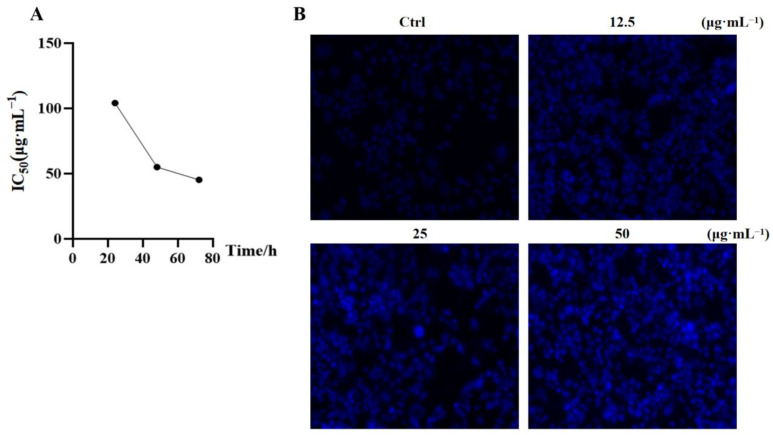
PEIM changed the growth and morphology of HepG2 cells. (**A**) The effect of PEIM on the viability of HepG2 cells was determined by MTT assay. (**B**) Apoptotic morphology of HepG2 cells after treatment with PEIM. Cells were stained with Hoechst 33342 solution for 15 min at room temperature (The magnification is 10×).

**Figure 4 pharmaceuticals-17-00806-f004:**
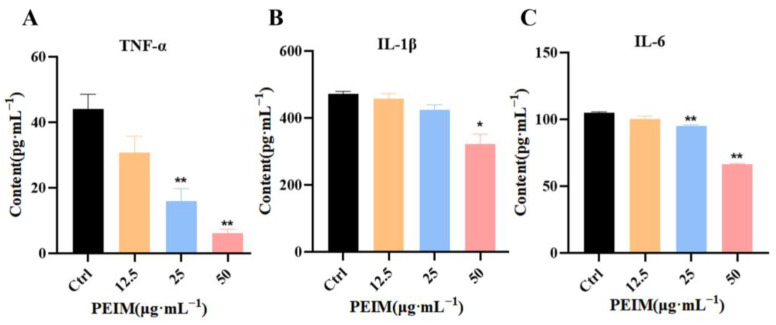
PEIM reduced the content of inflammatory factors. (**A**) The content of TNF-α after treatment with PEIM, ** *p* < 0.01 compared to the control group. (**B**) The content of IL-1β after treatment with PEIM, * *p* < 0.05 compared to the control group. (**C**) The content of IL-6 after treatment with PEIM, ** *p* < 0.01 compared to the control group.

**Figure 5 pharmaceuticals-17-00806-f005:**
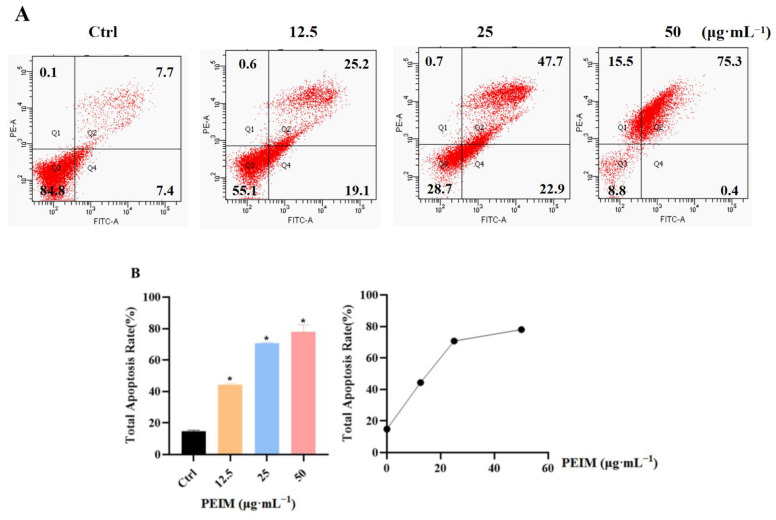
PEIM promoted apoptosis of HepG2 cells. (**A**) The apoptosis of HepG2 cells detected by flow cytometer after treatment with PEIM. (**B**) The apoptosis rate of HepG2 cells detected by flow cytometer after treatment with PEIM, * *p* < 0.05 compared to the control group.

**Figure 6 pharmaceuticals-17-00806-f006:**
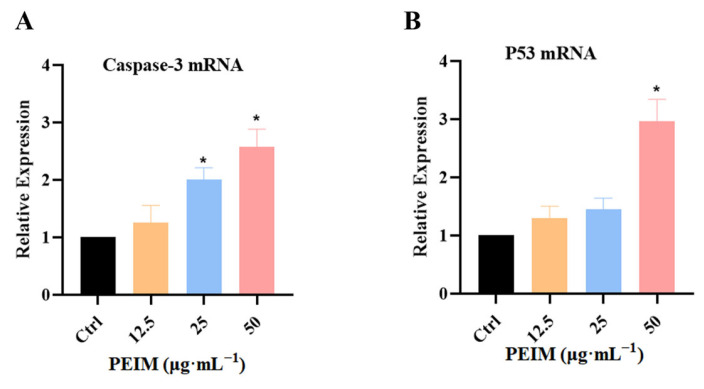
PEIM affected the expression of P53 mRNA and caspase-3 mRNA in HepG2 cells. (**A**) PEIM promoted the expression of P53 mRNA in HepG2 cells, * *p* < 0.05 compared to the control group. (**B**) PEIM promoted the expression of caspase-3 mRNA in HepG2 cells, * *p* < 0.05 compared to the control group.

**Figure 7 pharmaceuticals-17-00806-f007:**
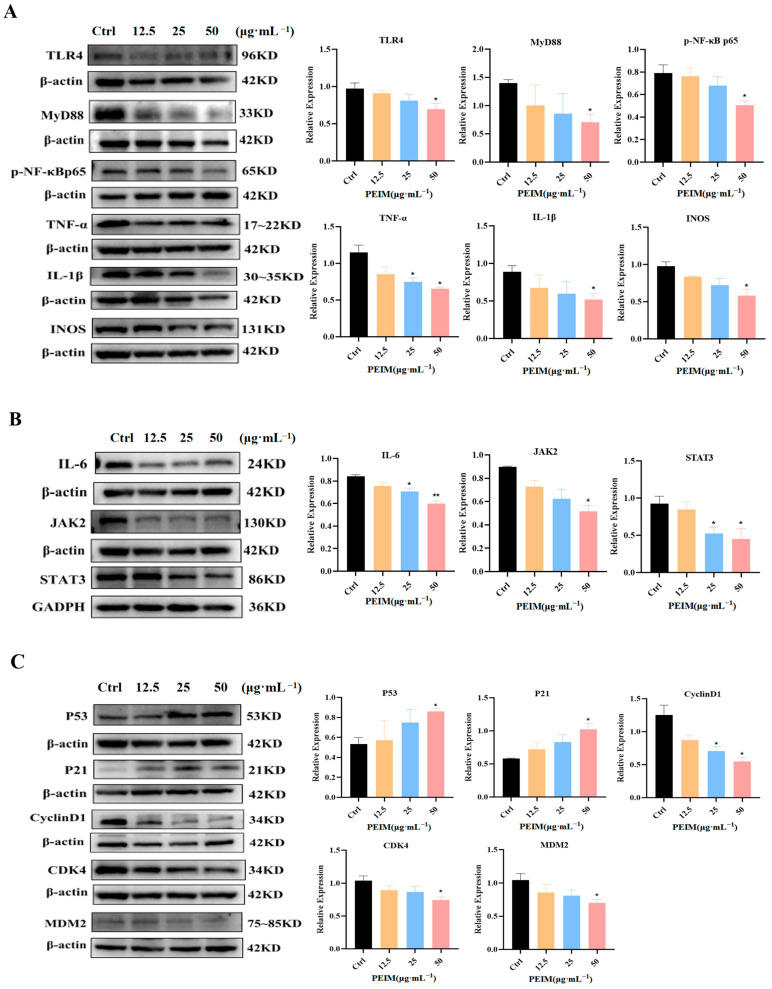
PEIM regulated the expression of TLR4/MyD88/NF-κB, JAK2/STAT3, P53/P21/MDM2, and mitochondrial apoptosis pathway-related proteins. (**A**) Expression of proteins involved TLR4/MyD88/NF-κB signaling pathway of HepG cells treated with PEIM. * *p* < 0.05 compared to the control group. (**B**) Expression of proteins involved JAK2/STAT3 signaling pathway of HepG cells treated with PEIM. * *p* < 0.05, ** *p* < 0.01 compared to the control group. (**C**) Expression of proteins involved P53/P21/MDM2 signaling pathway of HepG cells treated with PEIM. * *p* < 0.05 compared to the control group. (**D**) Expression of proteins involved mitochondrial apoptosis signaling pathway of HepG cells treated with PEIM. * *p* < 0.05, ** *p* < 0.01 compared to the control group.

**Figure 8 pharmaceuticals-17-00806-f008:**
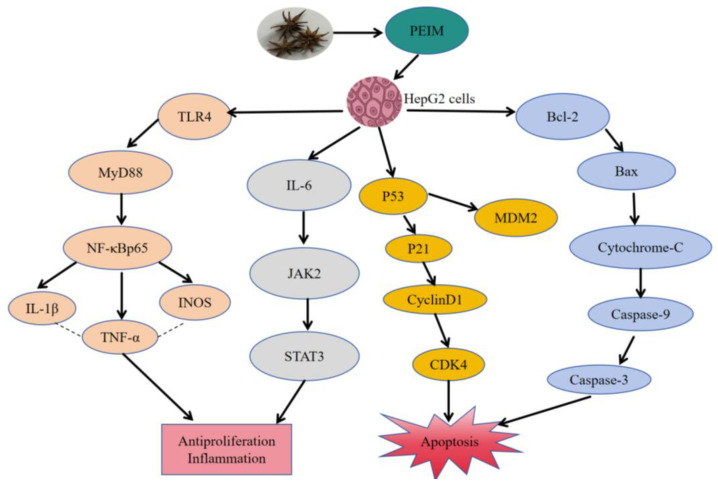
Mechanism of PEIM-induced HepG2 cells apoptosis and antiproliferation.

**Table 1 pharmaceuticals-17-00806-t001:** Compounds identified by LC-MS/MS in the PEIM.

NO.	RT (min.)	Precursor *m*/*z*	Fragmentation	Molecular Formula	Possible Compounds Identified
		[M+H]^+^ [M−H]^−^			
Flavonoids					
1	11.36	355.1180	355.1201, 135.0449	C_20_H_18_O_6_	Erythrinin C
2	14.05	325.1432	131.0511, 103.0567	C_20_H_20_O_4_	Glabridin
3	9.39	323.1278	323.1294, 161.0605, 135.0456	C_20_H_18_O_4_	Glabrene
4	5.76	515.1910	515.1897	C_27_H_30_O_10_	Baohuoside I
5	7.69	287.0552	287.0571, 153.0202	C_15_H_10_O_6_	Kaempferol
6	14.05	325.1434	131.0511, 103.0567	C_20_H_20_O_4_	Isobavachin
7	7.90	317.0660	317.0658, 302.0441, 153.0203,	C_16_H_12_O_7_	Isorhamnetin
8	6.62	303.0500	303.0498, 229.0508, 153.0200	C_15_H_10_O_7_	Morin hydrate
9	7.21	577.1562	369.0970, 346.9870, 297.1123, 108.0188	C_27_H_30_O_14_	Kaempferitrin
10	9.80	269.0820	269.1349, 225.0925, 149.0614	C_16_H_14_O_4_	Alpinetin
11	5.01	593.1510	593.1449, 285.0322	C_27_H_30_O_15_	Aempferol-3-O-rutinoside
12	4.52	609.1464	609.1445, 300.0280, 255.0291, 179.0019, 151.0070	C_27_H_30_O_16_	Kaempferol-3-gentiobioside
13	7.14	371.1140	371.1179, 327.1230, 297.1129, 267.0613, 160.0536	C_20_H_20_O_7_	Isosinensetin
14	6.61	301.0351	301.0355, 178.9978, 151.0025, 107.0136	C_15_H_10_O_7_	Quercetin
15	7.54	269.0458	269.0475, 228.9901, 151.0048, 117.0352	C_15_H_10_O_5_	Genistein
16	10.48	313.0719	313.1500, 283.0253, 255.0330, 216.9912, 145.0307	C_17_H_14_O_6_	Pectolinarigenin
17	5.13	623.1620	623.1631, 315.0606, 299.0172, 259.0542	C_28_H_32_O_16_	Isorhamnetin-3-O-neohespeidoside
18	5.19	447.0932	447.0894, 284.0345, 227.0324, 174.9554, 146.9589	C_21_H_20_O_11_	Quercitrin
19	7.67	285.0401	285.0390, 257.0452, 229.0486, 151.0040	C_15_H_10_O_6_	Luteolin
20	7.14	371.1139	371.1179, 327.1230, 282.0891, 267.0653, 160.0536	C_20_H_20_O_7_	Tangeretin
21	9.93	283.0610	283.0603, 268.0371, 242.9949, 152.9947	C_16_H_12_O_5_	Calycosin
22	4.70	463.0881	463.0875, 300.0278, 271.0220, 174.9579	C_21_H_20_O_12_	Hyperin
Diterpenoids					
23	6.60	557.1960	111.3465, 81.0746	C_36_H_28_O_6_	Neoprzewaquine A
24	16.58	351.2170	351.1984, 207.1384, 161.1346, 105.0723	C_20_H_30_O_5_	Andrographolide
25	17.49	533.2384	533.3047, 356.1428, 135.1185, 107.0874	C_28_H_36_O_10_	Butanedioicacid
26	12.86	297.1487	297.1535, 279.2671, 256.1101, 227.0734	C_19_H_20_O_3_	Cryptotanshinone
27	12.93	331.1900	313.2739, 271.2097, 149.0977, 133.1014	C_20_H_26_O_4_	Carnosol
28	12.13	333.2059	333.2167, 315.1934, 269.1915, 119.0892	C_20_H_28_O_4_	Carnosic acid
29	7.58	357.1342	357.1323, 342.1085, 232.9816, 137.0604, 83.0139	C_20_H_22_O_6_	Triptonide
30	6.11	359.1496	326.1166, 300.1245, 269.0821, 208.0739, 180.0745	C_20_H_24_O_6_	Triptolide
Triterpenoids					
31	16.78	439.3571	439.3638, 191.1792, 135.1175	C_30_H_46_O_2_	Ganoderiol A
32	19.09	455.3521	455.3567, 437.3455, 237.2719, 161.1372	C_30_H_46_O_3_	Betulonicacid
33	12.82	537.2980	537.3027	C_30_H_4l_O_6_	Senegenin
34	12.82	517.3166	517.3328, 365.2006, 347.1898	C_30_H_46_O_7_	Ganoderic acid C2
35	16.73	467.3167	467.3342, 423.3413, 399.1739, 125.0998	C_30_H_44_O_4_	Ganoderic acid DM
36	13.76	471.3478	471.3559	C_30_H_48_O_4_	Ganodermanontriol
37	16.77	469.3323	469.3325, 451.3229, 425.3417	C_30_H_46_O_4_	Glycyrrhetinic Acid
38	16.77	455.3530	455.3522, 407.3415, 155.0367	C_30_H_48_O_3_	Betulinic acid
39	11.28	487.3430	487.3406, 229.0074	C_30_H_48_O_5_	Asiatic acid
Alkaloids					
40	7.07	308.2221	308.2257, 290.2117, 136.0786, 122.0184	C_18_H_29_NO_3_	Dihydrocapsaicin
41	14.16	213.1021	213.1322, 183.0993, 172.0899, 129.0718	C_13_H_12_N_2_O	Harmine
42	0.61	272.1282	148.9876, 126.0561, 108.0458	C_16_H_17_NO_3_	Higenamine
43	13.66	457.2334	457.2371, 123.1173	C_25_H_32_N_2_O_6_	Vindoline
44	20.19	623.3131	623.3101, 563.2927	C_38_H_42_N_2_O_6_	Tetrandrine
45	3.38	286.1440	286.1455, 256.0915, 226.0412	C_17_H_19_NO_3_	Piperine
46	4.13	326.1595	326.1576, 182.9536	C_16_H_23_NO_6_	Monocrotaline
47	8.71	286.1082	286.0996, 196.0750, 168.0871, 107.0369	C_16_H_17_NO_4_	Lycorine
Lignans					
48	9.81	403.2116	403.2069, 129.0178	C_23_H_30_O_6_	Schisanhenol
49	6.12	539.2280	367.1435, 343.1610, 163.0772	C_30_H_34_O_9_	Schisantherin E
50	9.52	343.1540	343.1588, 311.1283, 265.0868, 161.0607	C_20_H_22_O_5_	Arisantetralone A
51	11.36	355.1175	355.1201, 337.1097, 135.0449	C_20_H_18_O_6_	Asarinin
52	14.68	399.1086	399.1835, 381.1745	C_21_H_20_O_8_	4′-Demethylepipodophyllotoxin
53	11.77	383.1500	382.9986, 363.0072, 322.9807, 302.9933, 121.0280	C_22_H_24_O_6_	Schisandrin C
54	9.70	535.1969	535.1869, 341.1375, 193.0507, 134.0367	C_30_H_32_O_9_	Schisantherin A
Sesquiterpenoids					
55	13.53	285.1768	285.2242, 125.0979, 107.0864, 81.0711	C_15_H_24_O_5_	Dihydroartemisinin
56	8.78	233.1536	233.1522, 175.1139, 147.1192	C_15_H_20_O_2_	Alantolactone
57	8.36	251.1641	251.1280, 147.1217, 95.0863	C_15_H_22_O_3_	Nardosinone
58	12.67	237.1850	237.2203, 201.1652, 149.1349, 71.0517	C_15_H_24_O_2_	Curdione
59	13.29	235.1694	235.1711, 179.1091, 57.0713	C_15_H_22_O_2_	Curcumenol
Others:					
60	7.69	287.0552	287.0571, 153.0202	C_15_H_10_O_6_	3-Hydroxymorindone
61	9.76	177.0545	149.0246, 65.0400	C_10_H_8_O_3_	Hymecromone
62	4.27	365.1443	365.1316	C_15_H_24_O_10_	Harpagide
63	20.42	401.3775	401.2042, 360.1636, 331.1258	C_28_H_48_O	Campesterol
64	14.05	337.1070	131.0511, 103.0567	C_20_H_16_O_5_	Psoralidin

**Table 2 pharmaceuticals-17-00806-t002:** Primer Sequences.

Gene	Sense	Antisense
P53	GCTTTCCACGACGGTGAC	GCTCGACGCTAGGATCTGAC
Caspase-3	AGAGCTGGACTGCGCTATTGAG	GAACCATGACCCGTCCCTTG

## Data Availability

Data is contained within the article.

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
