# Peer review of "Potential Molecular Mechanism of Illicium simonsii Maxim Petroleum Ether Fraction in the Treatment of Hepatocellular Carcinoma"

_pharmaceuticals, 2024, doi:10.3390/ph17060806_

Round 1

Reviewer 1 Report

Comments and Suggestions for Authors

The manuscript “Potential molecular mechanism of Illicium simonsii Maxim petroleum ether fraction in the treatment of hepatocellular carcinoma” aim to investigate the effect and mechanism of the petroleum ether fraction of Illicium simonsii Maxim on HepG2 cells. Overall, the manuscript is well written and easy to follow. Please find below comments which may be useful.

Major comments:

1.      I would recommend to include the LC-MS/MS spectra and method details.

2.      Please include the mass spectra of each compound in the supplementary data.

3.      Please discuss in detail on the previous studies on Illicium simonsii characterization.

4.      Please elaborate the conclusion section.

Keeping in view of above observations, I recommend a major revision of the manuscript.

Comments on the Quality of English Language

Minor editing of English language required

Reviewer 2 Report

Comments and Suggestions for Authors

The manuscript titled "Potential molecular mechanism of Illicium simonsii Maxim pentroleum ether fraction in the treatment of hepatocellular carcinoma," by Si-Hua Zou et al.  presents a comprehensive investigation into the effects of the petroleum ether fraction of Illicium simonsii Maxim (PEIM) on HepG2 cells, which is a significant contribution to the field of hepatocellular carcinoma (HCC) research. The identification of 65 compounds in PEIM using LC-MS/MS analysis is a robust approach that provides a clear chemical profile of the extract. The manuscript explores multiple signaling pathways (TLR4/MyD88/NF-κB, IL-6/JAK2/STAT3, P53/P21/MDM2, and mitochondrial apoptosis pathway) which adds depth to the understanding of PEIM's mechanism of action. Over all, the manuscript presents valuable findings that contribute to the understanding of potential therapeutic agents for HCC. With some revisions and additional details, this paper was potential to be published in this journal. Here are my comments.

1. HPLC analysis was encouraged to be incorporated to ascertain the relative content.

2. The results are presented clearly, but it would be helpful to have a summary figure that encapsulates the key findings for each pathway investigated.

The discussion should provide a more thorough analysis of the results in the context of existing literature. It should also address any unexpected findings or those that contrast with

3. The figures and tables are informative, but some may benefit from higher resolution or clearer labeling for better readability.

Reviewer 3 Report

Comments and Suggestions for Authors

Zou and colleagues prepared a manuscript on the Potential molecular mechanism of Illicium simonsii Maxim petroleum ether fraction in the treatment of hepatocellular carcinoma. The manuscript makes an extremely unfavorable impression due to the carelessness with which the authors approached the preparation of the material they received. A huge number of spelling, grammatical and linguistic errors are present throughout the entire text of the article. The abstract contains repeated conclusions, so, for example, the text on lines 16-17 repeats the text on lines 20-22. There are no explanations for many abbreviations. Line 29 – HCC?, Latin names should be written in italics (Lines 46-48, 49, etc.). Lines 47, 49, 56, 59, 71 also contain typos, etc. throughout the entire text of the article. As for the content of the introduction, discussion and conclusion, the entire text here has the character of fragmentary sentences that are not consistent with each other. The introduction does not justify the relevance of this study, is illogical and represents sentences taken out of context. The table also contains typos and inconsistencies. Thus, compound 24 is described twice as compound 29, but they have different molecular weights with the same brutto-formula and different percentages. Triterpenoids cannot in any way be triperpenoids, although in the table and further in the text of the article they are called that way. Figure 1 contains unreadable formulas of poor quality. The discussion contains more literary references than a discussion of the results obtained; there is no structure. The conclusions are accordingly very brief and also do not clarify the study. My opinion is that in its present form the manuscript cannot be published in this journal and only after extensive corrections and structuring of the results obtained can it be reconsidered for publication in the journal.

Round 2

Reviewer 1 Report

Comments and Suggestions for Authors

Authors have addressed all the raised comments and the manuscript can be accepted in its present form.

Reviewer 3 Report

Comments and Suggestions for Authors

In general, the authors took into account my wishes and comments and the manuscript can be published in the journal